# Comprehensive High-Depth Proteomic Analysis of Plasma Extracellular Vesicles Containing Preparations in Rett Syndrome

**DOI:** 10.3390/biomedicines12102172

**Published:** 2024-09-24

**Authors:** Sho Hagiwara, Tadashi Shiohama, Satoru Takahashi, Masaki Ishikawa, Yusuke Kawashima, Hironori Sato, Daisuke Sawada, Tomoko Uchida, Hideki Uchikawa, Hironobu Kobayashi, Megumi Shiota, Shin Nabatame, Keita Tsujimura, Hiromichi Hamada, Keiichiro Suzuki

**Affiliations:** 1Department of Pediatrics, Graduate School of Medicine, Chiba University, 1-8-1 Inohana, Chuo-ku, Chiba-shi 260-0856, Chiba, Japan; sagi19891022@yahoo.co.jp (S.H.); uchikawa@chiba-u.jp (H.U.);; 2Department of Pediatrics, Asahikawa Medical University, 2-1-1-1 Midorigaoka Higashi, Asahikawa City 078-8510, Hokkaido, Japan; satoru5p@asahikawa-med.ac.jp; 3Department of Applied Genomics, Kazusa DNA Research Institute, Kisarazu 292-0818, Chiba, Japan; mishika@kazusa.or.jp (M.I.);; 4Department of Pediatrics, Eastern Chiba Medical Center, Togane 283-8686, Chiba, Japan; 5Department of Pediatrics, Asahi General Hospital, 1326, I, Asahi 289-2511, Chiba, Japan; 6Department of Pediatrics, Tokyo Women’s Medical University Yachiyo Medical Center, 477-96, Oowadashinden, Yachiyo City 276-8524, Chiba, Japan; 7Department of Pediatrics, Osaka University Graduate School of Medicine, Suita 565-0871, Osaka, Japan; 8Group of Brain Function and Development, Nagoya University Neuroscience Institute of the Graduate School of Science, Nagoya 464-8602, Aichi, Japan; 9Research Unit for Developmental Disorders, Institute for Advanced Research, Nagoya University, Nagoya 464-0804, Aichi, Japan; 10Institute for Advanced Co-Creation Studies, Osaka University, 1-3 Machikaneyama, Toyonaka 560-8531, Osaka, Japan; 11Graduate School of Engineering Science, Osaka University, 1-3 Machikaneyama, Toyonaka 560-8531, Osaka, Japan; 12Graduate School of Frontier Bioscience, Osaka University, 1-3 Yamadaoka, Suita 565-0871, Osaka, Japan

**Keywords:** Rett syndrome, extracellular vesicles containing preparations, high-depth proteome analysis, PI3K/AKT/mTOR signaling pathway, UBE3B

## Abstract

**Backgroud:** Rett syndrome is a neurodevelopmental disorder that affects 1 in 10,000 females. Various treatments have been explored; however, no effective treatments have been reported to date, except for trofinetide, a synthetic analog of glycine-proline-glutamic acid, which was approved by the FDA in 2023. Serological biomarkers that correlate with the disease status of RTT are needed to promote early diagnosis and to develop novel agents. **Methods:** In this study, we performed a high-depth proteomic analysis of extracellular vesicles containing preparations extracted from patient plasma samples to identify novel biomarkers. **Results:** We identified 33 upregulated and 17 downregulated candidate proteins among a total of 4273 proteins in RTT compared to the healthy controls. Among these, UBE3B was predominantly increased in patients with Rett syndrome and exhibited a strong correlation with the clinical severity score, indicating the severity of the disease. **Conclusions:** We demonstrated that the proteomics of high-depth extracellular vesicles containing preparations in rare diseases could be valuable in identifying new disease biomarkers and understanding their pathophysiology.

## 1. Introduction

Rett syndrome (RTT) is a neurodevelopmental disorder affecting 1 in 10000 live female births [1]. Patients normally develop RTT within 6 to 18 months of age, after which motor function and communication skills begin to decline, causing repetitive hand movements, such as wringing, clapping, and washing. Patients may experience a loss of coordinated hand movements, which can progress to severe physical dysfunction and necessitate wheelchair dependence in some cases [2]. Complications of RTT include microcephaly in 80% of cases [3], gait disturbance in 80% [4], epilepsy in 60.8% [5], and scoliosis in 85%, with 13% of these cases requiring surgery [6,7]. Abdominal disorders, such as gastroesophageal reflux and air swallowing with abdominal distention, are also common and sometimes serious [8]. Autonomic symptoms, such as breathing dysregulation characterized by both hyperventilation and breath holding, as well as the dysregulation of temperature in the extremities, have also been identified as specific findings of RTT [9].

*Methyl-CpG-binding protein2* (*MECP2*) gene abnormality on chromosome Xq28 has been reported as a typical causative factor for RTT, being reported in approximately 90% of patients [10,11,12]. The Human Gene Mutation database identifies 555 *MECP2* mutations associated with RTT; however, 70% of patients with RTT with *MECP2* gene abnormalities have mutations in R106W, R133C, T158M, R168X, R255X, R270X, R294X, and R306C [12,13]. *MECP2* variations have not been identified in approximately 3–5% of cases meeting the clinical diagnostic criteria for RTT [12]. Additionally, these variations have been reported in autism [14] and Angelman syndrome [15]. Therefore, the presence of an *MECP2* mutation does not definitively indicate RTT. In order to diagnose typical or classic RTT, it is necessary to observe the following: (1) a period of regression followed by recovery or stabilization, (2) all main criteria and all exclusion criteria, And (3) supportive criteria, which are not required, although often present in typical RTT. The main criteria are as follows: (1) partial or complete loss of acquired purposeful hand skills, (2) partial or complete loss of acquired spoken language, (3) gait abnormalities, including impaired ability (dyspraxic) or the absence of ability, And (4) stereotypic hand movements, such as hand wringing/squeezing, clapping/tapping, mouthing and washing/rubbing automatisms. The exclusion criteria are as follows: (1) a brain injury secondary to trauma (peri- or postnatally), neurometabolic disease, or severe infection that causes neurological problems and (2) grossly abnormal psychomotor development in the first 6 months of life [9]. It has been documented that 70% of patients diagnosed with typical or classic RTT survive to the age of 45 [16].

Various drugs and treatments have been tested for RTT, including ketogenic diets [17], naltrexone [18], omega-3 fatty acids [19,20], triheptanoin [21], folinic acid [22,23,24], lovastatin [25], cerebrolysine [26], fingolimod [27], and IGF-1 [28,29]. However, no effective treatments have been reported to date, except for trofinetide, a synthetic analog of glycine-proline-glutamic acid, which was approved by the FDA for patients suffering from RTT [30,31].

We believe that there are treatments to prevent and treat RTT symptoms. This belief is based on the observation that, while increased cell packing density and decreased dendrites occur in the neurons of patients with RTT, neuronal loss or apparent gliosis is not observed [32]. In an experiment using conditional knock-in mice, *MECP2* function was restored in Floxed Stop/Y mice, which developed RTT-like symptoms at approximately 4–6 weeks and died around 10 weeks. However, tamoxifen alleviated neurological and respiratory symptoms and extended their lifespan to 30 weeks or longer [33]. The loss of MECP2 protein did not lead to cytotoxicity, suggesting that the restoration of *MECP2* gene function or the replacement of MECP2 protein could potentially alleviate the symptoms of RTT.

As mentioned above, RTT can only be diagnosed clinically; that is, it can only be diagnosed after the symptoms have progressed to a certain degree. Therefore, significant attempts have been made to identify the biomarkers for early treatment. Various EEG parameters [23,29,34,35], plethysmographic parameters, autonomic indices, and blood-based oxidative stress markers have been reported as possible biomarkers [19,29,36]. However, most of them remain secondary or exploratory indicators owing to their unknown functional significance. Thus far, no effective biomarkers have been established to evaluate disease status.

A proteomic analysis using fibroblasts from patients with RTT and healthy controls has provided insights into protein alterations related to the mitochondrial network [37]. Therefore, we performed a proteomic analysis of extracellular vesicles containing preparations (EVs-cp) in the plasma of patients with RTT to identify proteins that could serve as new biomarkers. Extracellular vesicles are defined as particles that are released from cells, that are delimited by a lipid bilayer, and that cannot replicate on their own according to MISEV (Minimal Information for Studies of Extracellular Vesicles) guidelines “https://www.isev.org/misev (accessed on 24 March 2024)”. Extracellular vesicles can cross the cerebrovascular barrier and are considered potential biomarkers for central nervous system diseases. The rationale for focusing on extracellular vesicles is that, recently, studies focused on identifying new biomarkers by analyzing extracellular vesicles, including exosomes, have been successfully conducted [38]. It was postulated that a comparable investigation of extracellular vesicles or EVs-cp in Rett syndrome might yield a novel biomarker. Using high-depth data-independent acquisition (DIA), we successfully identified over 2000 proteins. Subsequently, a novel candidate biomarker protein, UBE3B, was identified.

## 2. Materials and Methods

### 2.1. Subjects and Collection of Plasma Samples

A total of six patients who met the diagnostic criteria for typical Rett syndrome and had a *MECP2* gene mutation confirmed by genetic testing were selected from the Chiba University Hospital patient population. All patients with Rett syndrome exhibited a de novo onset, and the mutation was heterozygous. The Clinical Severity Score (CSS) was utilized to assess the severity of the patient’s condition. The CSS is a 13-item scale that assigns a score based on the timing of regression and the extent of clinical findings. The CSS is expressed as a total score ranging from 1 to 58, with higher scores indicating greater severity. A detailed account of the scoring for each patient is provided in Appendix A.

All patients with RTT and neurotypical developmental controls (NCs) were enrolled at Chiba University Hospital after obtaining written informed consent for the current study. The clinical characteristics of the patients are listed in Table 1 and Appendix A. The age and sex of the patients with RTT and NCs were also matched. Blood samples were taken from six patients with RTT with clinically and genetically confirmed diagnoses (Table 1 and Appendix A) and six NCs. Samples were drawn using RNA Complete BCT (Streck, La Vista, NE, USA). The samples were centrifuged at 1800× *g* at room temperature for 15 min. The collected supernatant was then centrifuged again at 2800× *g* at room temperature for 15 min and immediately stored at −80 °C for future analysis.

### 2.2. Sample Preparation for Proteome Analyses

#### 2.2.1. Purification of EVs-Cp from Human Plasma

Human plasma EVs-cp were purified using an immunoprecipitation method combined with an ExoCap Streptavidin Kit (MBL Co., Ltd., Tokyo, Japan) and Maelstrom 8 Autostage (Taiwan Advanced Nanotech Inc., Taoyuan, Taiwan). Subsequently, 250 µL of human plasma was centrifuged at 15,000× *g* and 4 °C for 30 min. The supernatants (150 µL) were then transferred to separate tubes. Treatment buffer (150 µL, provided with the kit) was added to the supernatants and gently mixed. For immunoprecipitation, 1 mg/mL biotinylated anti-CD9 mAb (MBL Co., Ltd., Tokyo, Japan), anti-CD63 mAb (MBL Co., Ltd., Tokyo, Japan), and anti-CD81 mAb (MBL Co., Ltd., Tokyo, Japan) were mixed (1:1:1, *v*/*v*/*v*). Subsequently, 80 µL of streptavidin magnetic beads (Sera Mag SpeedBeads Blocked Streptavidin particles, 80 µg solids, Cytiva, Marlborough, MA, USA) was washed with 300 µL of Treatment Buffer. After washing, 2.5 µL of the mAb mixture was added to the beads in 200 µL of Treatment Buffer and incubated for 10 min. The beads were then washed twice with 200 µL of Treatment Buffer before adding to the samples. To bind EVs-cp to the beads, the mixture was agitated at 1000 rpm for 2 h, followed by washing thrice with 500 µL of Washing/Dilution Buffer (provided with the kit). Finally, bead-captured EVs-cp were eluted with 100 µL of 100 mM Tris-HCl (pH 8.0) and 20 mM of NaCl containing 4% SDS.

#### 2.2.2. Sample Preparation for Proteome Analysis

The protein extracts were reduced with 20 mM tris (2-carboxyethyl) phosphine for 10 min at 80 °C, followed by alkylation with 30 mM iodoacetamide at room temperature for 30 min in the dark. Protein purification and digestion were performed using sample preparation (SP3) method [39,40]. Tryptic digestion was performed using 500 ng/µL Trypsin/Lys-C Mix (Promega, Madison, WI, USA) overnight at 37 °C. The digests were purified using GL-Tip SDB (GL Sciences, Tokyo, Japan) according to the manufacturer’s protocol. Peptides were dissolved in 2% acetonitrile (ACN) containing 0.1% TFA.

### 2.3. LC-MS/MS

The digested peptides were loaded directly using a C18 nano-capillary column (IonOpticks, Fitzroy, VIC, Australia) with dimensions of 75 µm × 25 cm at 60 °C. Then, the peptides were separated using a 120 min gradient with mobile phase A (0.1% FA in water) and mobile phase B (0.1% FA in 80% ACN). The gradient consisted of 7% B at 0 min and 65% B at 120 min, with a flow rate of 200 nL/min, using an UltiMate 3000 RSLCnano LC system (Thermo Fisher Scientific, Waltham, MA, USA). The eluted peptides were detected using a quadrupole Orbitrap Exploris 480 hybrid mass spectrometer (Thermo Fisher Scientific, Waltham, MA, USA) with a normal DIA window. The MS1 scan range was set as a full scan with *m/z* values in the range of 475–845 at a mass resolution of 15,000. The Auto Gain Control (AGC) target for MS1 was set to 3 × 10^6^ with the maximum injection time set to “auto”. The MS2 were collected for *m/z* values greater than 200 at 60,000 resolutions, with an AGC target of 3 × 10^6^, the maximum injection time set to “auto”, and stepped normalized collision energy values of 22, 26, and 30%. The isolation width for MS2 was set to 6 *Th*, and for the *m/z* range of 475–845, an optimized window arrangement was used in Scaffold DIA (Proteome Software, Inc., Portland, OR, USA).

### 2.4. Data Processing

Raw data were searched against an in silico-predicted spectral library using DIA-NN [41], version:1.8.1, “https://github.com/vdemichev/DiaNN (accessed on 1 March 2023)”. First, the in silico-predicted spectral library was generated from the Human UniProtKB/Swiss-Prot database (UniProt id UPUP000005640, 20588, reviewed, canonical, UP000005640, 20588 entries, 11 November 2021) using DIA-NN. The DIA-NN search parameters were as follows: protease, trypsin; missed cleavages, 1; peptide length range, 7–45; precursor charge range, 2–4; precursor mass range, 480–840; fragment ion m/z range, 200–1800; mass accuracy, 10 ppm; static modification, cysteine carbamidomethylation; enabled “Heuristic protein interferences”, “use isotopologues”, “MBR”, and “no shared spectra”. Additional commands were set as follows: “mass acc cal 10”, “peak translation”, and “matrix specq”. The protein identification threshold was set to < 1% for both peptide and protein false discovery rates (FDRs). Statistical calculations and Pearson correlation coefficient heatmap analysis with hierarchical clustering were performed using Perseus v2.011 [42].

## 3. Results

A total of 4273 proteins were identified by performing an in-depth DIA proteome analysis of plasma collected from 12 participants (six patients with RTT and six NCs) (Table 1, Appendix A). An enrichment analysis of the identified proteins revealed extracellular vesicles containing proteins, confirming successful extraction (Figure 1A and Appendix A). The proteins were analyzed using Perseus ver. 2.011, and 2712 were selected based on the criteria that they were detected with two or more unique peptides in over 70% of the samples from either the RTT group or NC group.

We investigated the degree of overlap between the proteins extracted in our experiments, those identified by the CS-based isolation method most extensively documented by Lin Cao et al. [43], and those in one of the most reliable and utilized databases of EV, namely the Vesiclepedia database, version 5.1, “http://microvesicles.org/index.html (accessed on 5 September 2024)” [44]. From the protein/mRNA data in the Vesiclepedia database, we extracted only those entries corresponding to humans as the animal species and protein as the molecular species. The extracted proteins were further extracted from the DAVID database, version 2024q2, “https://david.ncifcrf.gov/home.jsp (accessed on 5 September 2024)”. A comparison was conducted between the proteins extracted in the present study, those extracted by Lin Cao et al. [43], and those listed in the Vesiclepedia database [44] (Figure 1B). A total of 77.5% of the extracted proteins in Lin Cao et al.’s study [43] were matched with the present study, while 3777 proteins (representing 88% of the total) were matched with the Vesiclepedia database [44]. This indicates that the extracted proteins in the present study are likely to contain extracellular vesicles. However, many proteins (452 proteins, representing 10.5% of the total) not listed in Vesiclepedia [44] were also detected compared to the results of the experiment by Lin Cao et al. [43] (31 proteins, representing 1.4% of the total). For this reason, we describe the proteins we extracted in this experiment as EVs-cp.

An F-test was performed, and no equal variances were found between the two groups. Therefore, the differences between the two groups were determined using Welch’s *t*-test (Appendix A).

The missing values were compensated using a width of 0.3 and a downshift of 2.8. Welch’s *t*-test was applied to the proteins of these two groups with Benjamini–Hochberg FDR correction (q = 0.05). The 33 upregulated proteins (red square) and 17 downregulated proteins (blue square) were defined as proteins with fold changes of 2 or more and less than 0.5, respectively (Figure 2 and Appendix A). The proteins that differed significantly between the two groups are shown in a heat map and dendrogram in Figure 3. Appendix A provide details on the *p*-values, Cohen’s D, molecular functions, and biological processes for each protein. The molecular function and biological process were cited from “Inter Pro Classification of protein families”, “https://www.ebi.ac.uk/interpro/ (accessed on 19 June 2024)” [45].

The upregulated proteins were analyzed using Shiny GO 0.80 “http://bioinformatics.sdstate.edu/go/ (accessed on 8 May 2024)” as follows: a false discovery rate (FDR) cutoff of q = 0.05, a pathway size of 2–5000, and the options “remove redundancy” and “abbreviate pathways” enabled. The enrichment analysis of the GO Biological Process is presented in Figure 4 and Appendix A. The highest fold enrichment was observed for the negative regulation of the amyloid precursor protein biosynthetic process. The proteins responsible for this are ITM2A and ITM2C. ITM2A was detected in the RTT group, except in RTT-5, whereas it was not detected in the NC group except in NC-4. ITM2C was detected in the RTT group, except in RTT-3, whereas it was not detected in the NC group except in NC-4.

The enrichment analysis of the KEGG pathway is shown in Figure 5 and Appendix A. Among the proteins involved in the PI3K-Akt signaling pathway, four were upregulated in the RTT group: COL4A2, ANGPT1, LAMB1, and LAMC1. Among these proteins, COL4A2 was detected in all patients in the RTT group, and only NC-2 and NC-4 were detected in the NC group. In addition, ANGPT1, LAMB1, and LAMC1 were detected in all patients.

We focus on the 33 up- and 17 downregulated proteins identified through a comparison between the RTT group and the NC group, and the correlation between the log2-transformed mass of the extracted proteins and the CSS in the RTT group was examined. A correlation coefficient (R) > 0.7 was defined as being indicative of a highly correlated relationship [46,47]. Twelve proteins, including nine from the upregulated group and three from the downregulated group, exhibited a strong correlation with the CSS (Figure 6 and Appendix A). Two of these proteins, UBE3B and CHL1, were expressed only in patients with RTT (Appendix A).

## 4. Discussion

Through a DIA proteomic analysis of plasma-isolated EVs-cp, we identified 33 upregulated and 17 downregulated proteins in participants with RTT compared to the NCs. Instead of a decrease in structural proteins, there was an increase in inflammation-associated pathways, such as ECM–receptor interaction and focal adhesion, including the PI3K/AKT signaling pathway. Among the 50 proteins analyzed, UBE3B, a ubiquitin ligase, exhibited the highest correlation coefficient with CSS in the RTT group. This finding suggests that UBE3B might serve as a novel biomarker for assessing the development or severity of RTT.

A number of studies have employed a proteomic analysis in the context of patients diagnosed with Rett syndrome [37,48,49]. To date, only two proteomic analyses using human serum have been reported. One of these compared two pairs of patients with Rett syndrome with the same mutation in sisters with healthy controls [48], while the other compared 25 patients with typical Rett syndrome with *MECP2* mutations with healthy controls [49]. In these studies, 6 and 17 significantly different proteins were identified, respectively. In the present study, 50 proteins were identified, representing a greater number than that reported in previous studies. Furthermore, while the majority of the previously reported proteins were elevated in inflammation-related proteins, several non-inflammatory proteins were identified in the present study. It is postulated that this discrepancy is attributable to the fact that the present study targets extracellular vesicles.

In current study, the PI3K/AKT/mTOR signaling pathway was upregulated in the RTT group. However, among the four related proteins, LAMB2, LAMC2, and COL4A2 were upregulated in the other five pathways. The PI3K/AKT pathway, a downstream signaling pathway of BDNF, is downregulated in patients with RTT [50]. However, RTT pathogenesis is thought to be associated with inflammation, and elevated levels of oxidative stress- and inflammation-related proteins have been reported [49,51,52]. It is possible that feedback at the expense of the decreased activity of the BDNF pathway may have resulted in a predominant increase in the proteins associated with these pathways.

In this study, we detected more than 2-fold significant differences in protein levels compared to the previously reported proteomic results. While previous reports have mainly highlighted inflammation-related proteins, our study identified DHX9, VNN1, and TBK1 as upregulated inflammation-related proteins and MMP25 as a downregulated one. DExH-box helicase 9 (DHX9), a member of the DExD/H-box RNA helicase superfamily II, is an autoantigen that induces serum inflammation in patients with systemic lupus erythematosus [53]. It also promotes macrophage-mediated inflammation [54] and interacts with the MECP2 protein [55]. VNN1 is a protein associated with oxidative stress, and it is overexpressed when oxidative stress is elevated [56]. We believe that oxidative stress is enhanced in RTT [57]. TBK1, which is rapidly activated in response to mitochondrial damage and is involved in autophagizing damaged mitochondria [58], is upregulated in Rett syndrome, suggesting abnormalities in mitochondrial function [59]. MMP25 is upregulated and downregulated when the PI3K/AKT/mTOR signaling pathway is activated and suppressed, respectively [60], although the underlying mechanism is unknown. There have been no reports of increased or decreased protein levels in patients with RTT.

While previous reports focused on the elevation of inflammation-related proteins, this study detected several other significantly different proteins, including inflammation-related proteins, which may be due to extracellular vesicle extraction.

In the present study, increased levels of plasma UBE3B were identified as a potential biomarker of RTT. On the other hand, some investigators reported that the level of BDNF—a neurotrophic factor—is decreased in patients with RTT [61,62]. The BDNF/TrkB pathway is involved in many brain functions, including neuronal survival, neurite outgrowth, and synapse formation [63]. BDNF cannot cross the BBB; therefore, the direct administration of BDNF has not shown therapeutic efficacy. However, trophinetide, an IGF-1 analog and a neurotrophic factor, can cross the BBB and mimic the effects of BDNF, resulting in symptomatic improvement [31]. Tyrosine phosphorylation residues outside the TrkB kinase activation domain Phospho-Tyr785 recruits and activates PLCγ, hydrolyzing phosphatidylinositol 4,5-bisphosphate to produce diacylglycerol (DAG) and inositol 1,4,5-trisphosphate (IP3) [53,64]. Reduced BDNF levels lead to decreased IP3 levels, which are inversely correlated with the level of the calcineurin catalytic subunit, PPP3CC [65]. In mouse studies, Ppp3cc is ubiquitinated by Ube3b, the mouse ortholog of UBE3B, in a manner that regulates correct spine density [66]. In conclusion, mutations in MECP2 reduce BDNF and IP3 levels, increase PPP3CC levels, and increase UBE3B levels. These proteins are involved in neuronal plasticity, suggesting that the severity of symptoms may correlate with their levels. If this hypothesis is correct, the symptoms of patients with RTT are likely due to the decreased expression of the neurotrophic factor BDNF. However, since BDNF cannot pass through the BBB, RTT cannot be treated directly with BDNF. The creation of a therapeutic drug that targets the downstream signaling protein may prove to be an effective means of improving symptoms. Furthermore, the assessment of treatments for Rett syndrome is primarily based on clinical observations. Given that the symptoms of Rett syndrome are associated with neurological degeneration caused by *MECP2* mutations, a long-term observational approach is necessary to monitor the healing process. Demonstrating a decline in UBE3B, a potential target protein, at the commencement of treatment could serve as a sensitive indicator for the evaluation of therapeutic efficacy.

Another protein, CHL1, which is exclusively found in patients with RTT, may be elevated based on the following hypothesis. The amyloid precursor protein (APP)—a protein produced by neurons and released from presynaptic terminals for use at dendritic synapses—is important for neurite outgrowth and synapse formation [67,68,69,70]. APP releases secreted metabolites upon cleavage. Degradation by the β-site APP cleavage enzyme (BACE1) via the amyloidogenic pathway results in the secretion of AICD and amyloid beta (Aβ), which is potentially associated with psychiatric symptoms and microcephaly in AD and ASD. In the non-amyloidogenic pathway, sAPPα is produced by α-secretase ADAM10 and cleaved by γ-secretase into AICD and P3 peptides. sAPPα and P3 are both neurotrophic and neuroprotective. MECP2 regulates ADAM10 via miRNA-197 [71], and *MECP2* expression and phosphorylation levels increase in the CA1 hippocampus of rats injected with Aβ [72]. The GO Biological Process with the greatest fold change in the enrichment analysis in this study was the Negative Regulation of Amyloid Precursor Protein Biosynthesis. This pathway is possibly enhanced by negative feedback owing to increased APP metabolites in patients with RTT (Figure 5). The related proteins, such as ITM2C (also known as BRI3) and BACE1, interact with each other, suggesting that the balance between ADAM10 and BACE1 is altered in patients with RTT [73,74]. ADAM10 and BACE1 activity regulate CHL1 levels in neurons and in vivo [75], with ADAM10 also promoting BACE1-mediated CHL1 proteolysis [76].

These findings suggest that CHL1 was expected to be detected in the RTT group. However, ADAM10 was found in plasma EVs-cp from both groups and did not show a significant difference between them. Similarly, L1CAM and CNTN2, whose intracellular concentrations were altered by ADAM10, BACE1, and CHL1, were either detected in only one patient with RTT or were not detected in any patient (Table 1). Whether these results reflect limitations in test sensitivity or underlying pathophysiological variations requires further investigation.

It should be noted that the present study is subject to a number of limitations. Firstly, the number of patient specimens collected in this study was relatively limited, with only six specimens being available for analysis. Our findings indicate that the protein UBE3B was predominantly expressed in the patient group. However, a further investigation is required to confirm whether similar results can be obtained in a large cohort of patients and whether the same test can be reproduced in a larger cohort of patients. Secondly, we did not perform a nanoparticle analysis and transmission electron microscopy on our prepared samples in the present study; therefore, the samples are of unknown particle size and are likely to contain non-vesicular entities. Meanwhile, in our samples, the EV markers such as CD9 and CD82 [77] showed high levels within the top 300 ranks of the identified proteins in all samples; these markers are generally found in low levels at under 1000 ranks in the plasma proteome analysis [78]. From these findings, we considered that EVs were collected in our samples and used the term “EVs-containing preparations” for our samples that may contain both EVs and non-vesicular entities according to the MISEV2023 recommendations [77]. Thirdly, 50 significantly differentially expressed proteins were identified (33 upregulated and 17 downregulated), representing more than double the number of proteins detected in the previous proteomics report. However, the function of these proteins could not be analyzed, so it will be necessary to verify whether these proteins are actually associated with MECP2 mutations. It is believed that only when UBE3B is found to be associated with MECP2 can it be expressed as a target protein for use as a biomarker.

## 5. Conclusions

Through DIA proteomics of EVs-cp samples extracted from the plasma of patients with RTT, we identified the upregulation of the PI3K/AKT/mTOR signaling pathway and the upregulation of UBE3B as potential biomarkers.

## Figures and Tables

**Figure 1 biomedicines-12-02172-f001:**
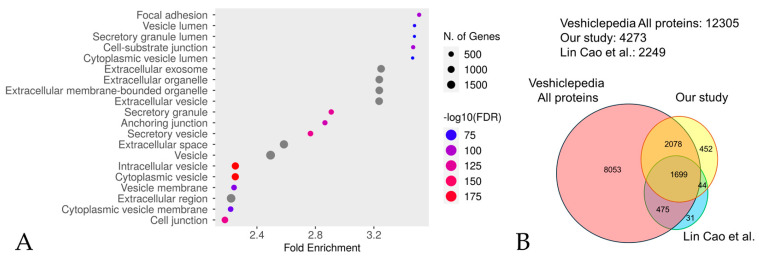
An enrichment analysis performed using Shiny GO 0.80 utilizing the GO Cellular Component database (**A**). The analysis was performed with the following settings: an FDR cutoff of q = 0.05, a pathway size range of 2 to 5000, and the options “remove redundancy” and “abbreviate pathways” enabled. The size of the circle represents the number of responsible genes, with larger circles representing more genes. The color represents the strength of the FDR, with red and blue indicating higher and lower FDR values, respectively. A comparison with Vesiclepedia (**B**). The vast majority (77.5%, 88%) of proteins identified in our study were also present in Lin Cao et al.’s study [43] and the Vesiclepedia database (version 5.1) [44], respectively.

**Figure 2 biomedicines-12-02172-f002:**
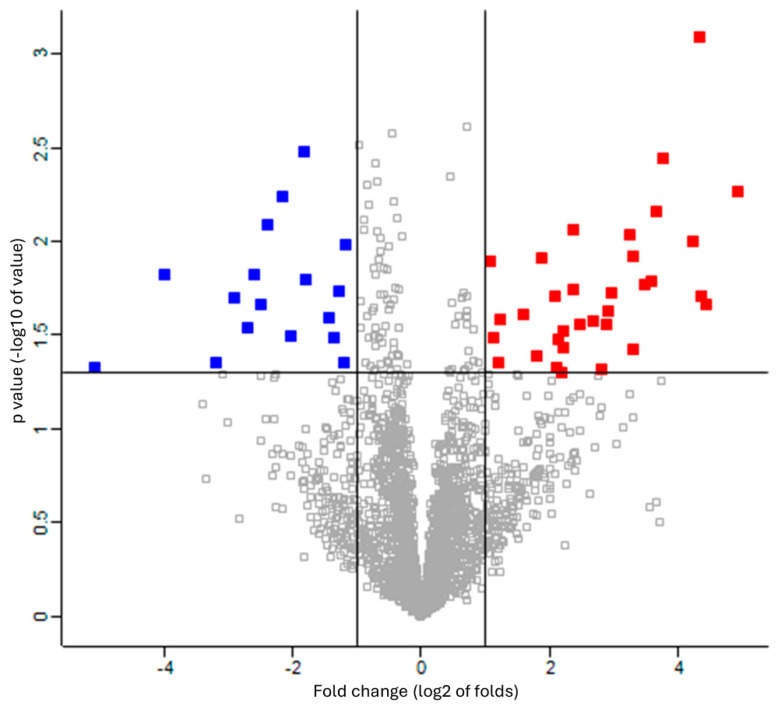
Volcano plot showing upregulation and downregulation of differentially expressed proteins among patients with RTT compared to NC group. Data points indicate different proteins that display both large-magnitude fold-changes (log2 of folds, *x*-axis) and high statistical significance (-log10 of *p* values, *y*-axis). Line on horizontal axis represents *p* = 0.05. Line on vertical axis represents fold changes of 2 and 0.5. Red and blue squares indicate proteins with fold changes of more than 2 and less than 0.5, respectively.

**Figure 3 biomedicines-12-02172-f003:**
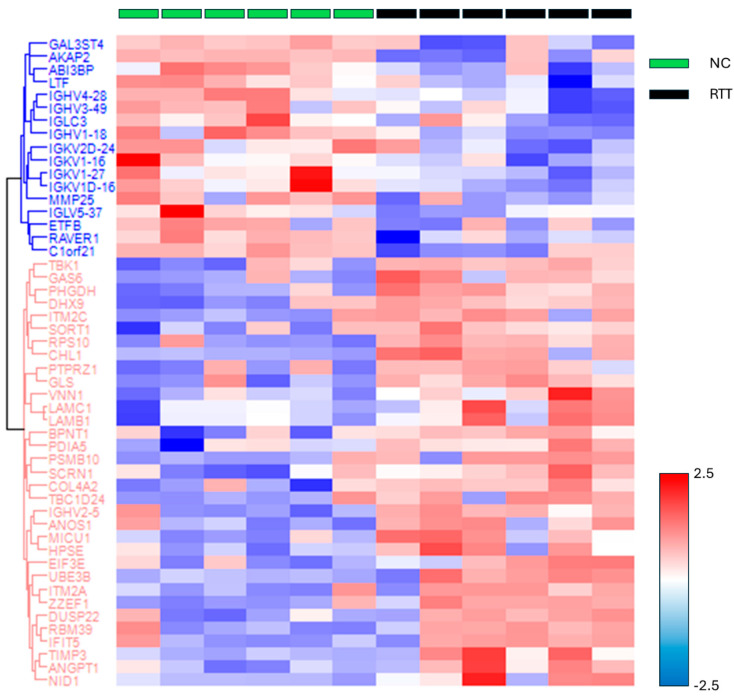
The heat map and dendrogram showing protein groups with significant differences between the RTT and NC groups. The log-transformed protein quantitative values were standardized using the z-score and color coded accordingly. Abbreviations: NC, neurotypical development controls; RTT, Rett syndrome.

**Figure 4 biomedicines-12-02172-f004:**
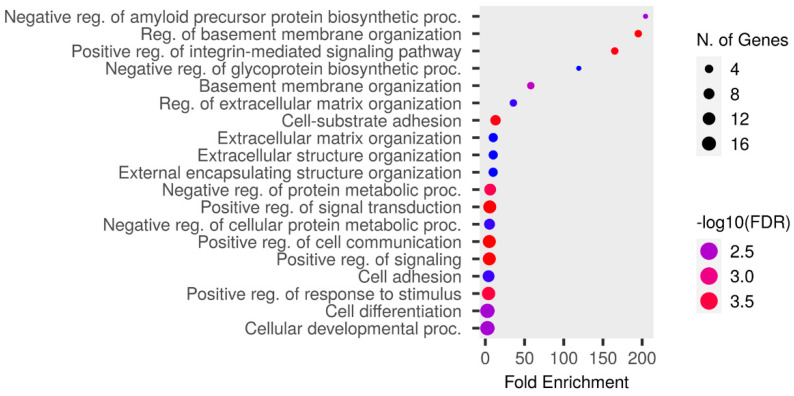
An enrichment analysis conducted using Shiny GO 0.80 utilizing GO Biological Process database. The analysis was performed under the following settings: an FDR cutoff of q = 0.05, a pathway size of 2–5000, and the options “remove redundancy” and “abbreviate pathways” enabled. The size of the circle represents the number of responsible genes, with a larger size indicating more genes. The color represents the strength of the FDR, with red and blue indicating higher and lower strengths, respectively.

**Figure 5 biomedicines-12-02172-f005:**
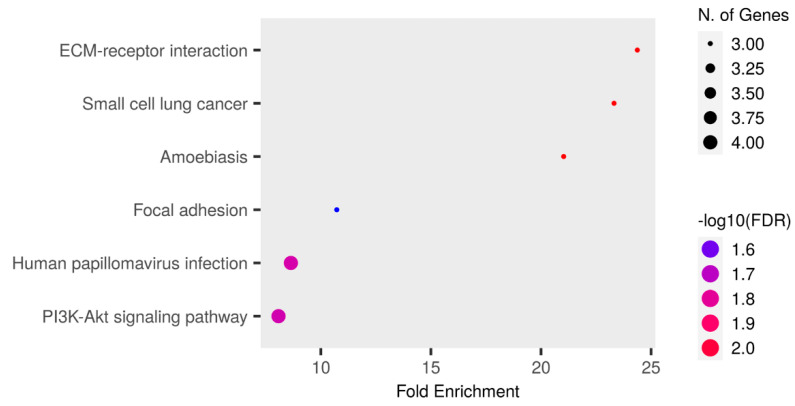
An enrichment analysis performed using Shiny GO 0.80. The Kegg Pathway database was used. The analysis was performed using the following settings: an FDR cutoff of q = 0.05, a pathway size of 2–5000, and the options “remove redundancy” and “abbreviate pathways” enabled. The size of the circle represents the number of responsible genes, with a larger circle indicating a greater number of genes. The color represents the strength of the FDR, with red and blue indicating higher and lower strengths, respectively.

**Figure 6 biomedicines-12-02172-f006:**
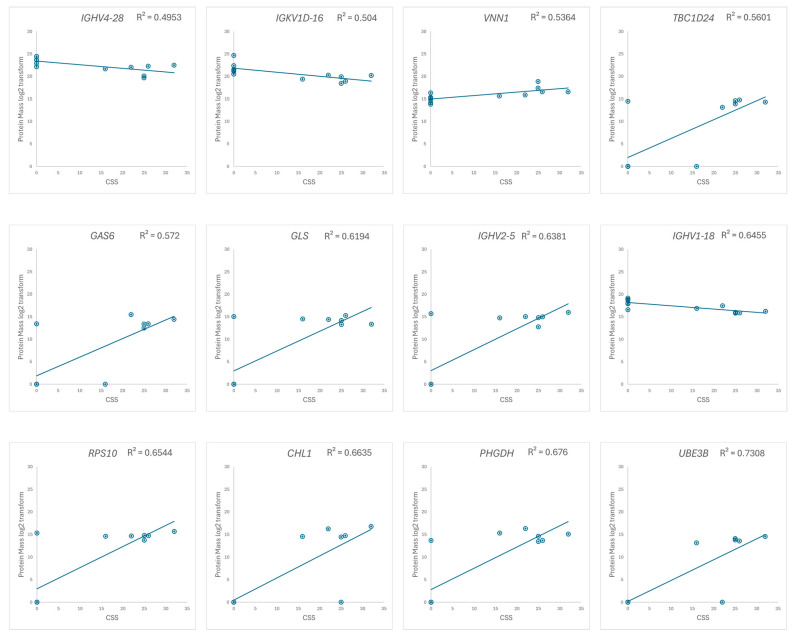
Scatter plots of the log2-transformed protein masses versus the CSS, with the regression line indicated. Proteins with correlation coefficients greater than 0.7 (R² > 0.49) are displayed. The vertical axis represents the protein mass, and the horizontal axis represents the CSS.

**Table 1 biomedicines-12-02172-t001:** Clinical background of patients with RTT.

		Patient 1	Patient 2	Patient 3	Patient 4	Patient 5	Patient 6
Age		9	15	8	4	7	22
Sex		Female	Female	Female	Female	Female	Female
Pathogenic variant in *MECP2*	nucleotide	c.806delG	c.763C > T	c.401C > T	c.401C > G	c.916C > T	c.316C > T
	protein	p.(Gly269Alafs*20)	p.(Arg255*)	p.(Ser134Phe)	p.(Ser134Cys)	p.(Arg306Cys)	p.(Arg106Trp)
CSS		25	26	25	16	22	32

## Data Availability

The original contributions presented in this study are included in the article/Appendix A; further inquiries can be directed to the corresponding author.

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
