# Peer review of "Comprehensive High-Depth Proteomic Analysis of Plasma Extracellular Vesicles Containing Preparations in Rett Syndrome"

_biomedicines, 2024, doi:10.3390/biomedicines12102172_

Round 1

Reviewer 1 Report

Comments and Suggestions for Authors

The authors presented the results of a high-resolution proteomic analysis of plasma exosomes in Rett syndrome. The study included six female patients (age range 4 -22 years) with a mutation in the MECP2 gene and 6 control patients. The study was designed to identify a potential therapeutic target for patients with Rett syndrome. While the aim of the study is of great importance, there are many points that need clarification and explanation.

1. Please use a consistent way of mentioning genes, in many cases in the main text, figures and supplementary materials, gene names are not written in italics.

2. I could not find the full name for CSS score. The abstract mentions Severity Scale Score without abbreviation, but in the article only CSS score is given. Please present the CSS score in some detail in the supplementary material.

3. In my opinion, clinical data are practically missing. Please include patient characteristics, clinical symptoms. Whether the mutations are new or relatives also have them, genetic information should be provided in more detail, including data on homo- or heterozygous status of the mutation. If relatives have the same mutations, please compare the severity of the disease. Compare disease severity by age. Please analyze differences between variants in disease severity using available online resources.

4. Indicate versions of bioinformatics tools used.

5. UBE3B protein level was downregulated in one of six patients (Fig.3), in another patient upregulation was weak. Please comment.

6. What is the main reason for choosing UBE3B as a target protein that it correlates most strongly with disease severity? However, the strong correlation depends on four people as shown in Figure 7 and the difference with the other proteins is small and the possible mechanism of interaction is unclear.  Although it is mentioned in the limitations, some speculations about the possible role of UBE3B in RTT syndrome should be presented, either for its role as a marker or as a participant in a causative interaction with MECP2. This is particularly important since UBE3B is not present in the supplementary materials with the results of enrichment analyses or in Figure 6 with protein-protein interactions. Furthermore, the major metabolic pathways and biological processes in which the UBE3B gene is involved are related to the immune system and ubiquitination.

7. Figure 6. “The minimum required interaction score was set to 0.15”. It is a weak level of evidence, the default level for medium level is 0.4. What is P-value for protein-protein interaction? Why are the results of this analysis given, what necessary information do they add?

8. In Figure 7, the axes are not signed.

9. There are two main conclusions in the article: the assumption that UBE3B can be considered as a potential target protein, which requires much deeper justification than given in the article, and the involvement of PI3K/AKT/mTOR in RTT syndrome, which is already known. As for UBE3B, it does not seem to be involved in the PI3K/AKT/mTOR pathway, so are the increased expression level of UBE3B and the upregulation of the PI3K/AKT/mTOR pathway two independent phenomena, or is there still some connection?

10. In the last sentence of the abstract, the authors write "We demonstrated that high-depth exosomal proteomics in rare diseases could be valuable in identifying new disease biomarkers and understanding their pathophysiology", but just to understand their pathophysiology, additional efforts and arguments should be made in the revision of the article.

Reviewer 2 Report

Comments and Suggestions for Authors

The manuscript presents a high-depth proteomic analysis of plasma exosomes from Rett syndrome (RTT) patients, aiming to identify novel biomarkers for early diagnosis and drug development. However, concerns regarding the experimental design, data interpretation, and manuscript structure need to be addressed before publication.

1. Adherence to the latest International Society for Extracellular Vesicles (ISEV) guidelines, particularly the 2023 Minimal Information for Studies of Extracellular Vesicles (MISEV) standards, is crucial for this exosome-focused study. The authors should ensure proper preliminary characterization of the exosomes and describe their methods clearly to meet these guidelines. Referring to the official MISEV guidelines available at (https://www.isev.org/misev) would provide specific recommendations.

2. The proteomic analysis presented only utilizes 12 biological samples, raising concerns about the robustness of the conclusions drawn. Additionally, the absence of experimental validation of the identified biomarkers undermines the credibility of the study. Additional experimental evidence, such as functional assays, is necessary to validate the significance of the proteomic findings. Without this, the claims made in the manuscript are insufficiently supported.

3. Although the abstract mentions the studys potential impact on drug development, the manuscript does not adequately explore this aspect. The authors should not only identify biomarkers but also discuss their potential implications for drug discovery and development, enhancing the manuscript's impact.

4. The discussion section lacks a thorough comparison with existing proteomic studies in RTT research. Engaging more comprehensively with prior studies to highlight how their findings contribute to or differ from existing knowledge would contextualize their work within the broader research landscape and strengthen their conclusions.

Comments on the Quality of English Language

Moderate editing of English language required.

Reviewer 3 Report

Comments and Suggestions for Authors

The manuscript is a valuable contribution to the field of Rett syndrome research. It addresses a critical need for biomarkers and provides a comprehensive proteomic analysis that could lead to significant advances in understanding the disease. The introduction could be improved by explicitly stating the study's hypothesis at the end. Additionally, a more in-depth discussion of previous proteomic studies in Rett syndrome would help to better position this research (cit 36 and related). Additionally, the potential clinical applications of these findings could be discussed in more detail, particularly how UBE3B might be used in a clinical setting.

Reviewer 4 Report

Comments and Suggestions for Authors

The RTT manuscript describes in great detail the research carried out and the results obtained, but some important information is missing. The following is a list of comments:

1. The general description of the disease should also include information on diagnosis. Is the diagnosis also based on genetic testing? Have there already been proteomics studies in this direction? What is the current state of knowledge on this subject?

2. What is the survival rate of the patients?

3. There should be a definition of exosomes in the manuscript and a justification of why it was decided to analyse them.

4. There is no development of the CSS score abbreviation

5. There should also be information in the summary about the limitations of the results obtained.

Round 2

Reviewer 1 Report

Comments and Suggestions for Authors

All my comments have been taken into account; I recommend acceptance of this work.

Author Response

Response to Reviewer 1 (Round 2)

All my comments have been taken into account; I recommend acceptance of this work.

Reply

Thank you so much for the peer review.

Reviewer 2 Report

Comments and Suggestions for Authors

The revisions made by the author to the paper have not significantly enhanced its quality, consequently rendering it unsuitable for publication in Biomedicines due to the following reasons:

 1. The analyses undertaken do not comply with the standards set by the International Society for Extracellular Vesicles. The absence of Western blot, transmission electron microscopy, and nanoparticle analysis is noteworthy.

 2. It is imperative for the authors to utilize western blot, ELISA, or immunofluorescence to validate the identified biomarkers.

Comments on the Quality of English Language

Moderate editing of English language required.

Round 3

Reviewer 2 Report

Comments and Suggestions for Authors

Despite some modifications to the description of extracellular vesicles, the author fails to provide compelling evidence that the obtained samples are indeed extracellular vesicles. This lack of proof extends to samples from other patients as well. As a result, the authenticity and reliability of the data in this paper remain dubious, and therefore it cannot be accepted for publication in Biomedicines.

Comments on the Quality of English Language

Moderate editing of English language required.
